# Blockchain Protocols and Edge Computing Targeting Industry 5.0 Needs

**DOI:** 10.3390/s23229174

**Published:** 2023-11-14

**Authors:** Miguel Oliveira, Sumit Chauhan, Filipe Pereira, Carlos Felgueiras, David Carvalho

**Affiliations:** 1Aveiro-North Polytechnic School, University of Aveiro, 3720-511 Oliveira de Azeméis, Portugal; 2Naoris Protocol, Wilmington, DE 19808, USAdavid@naoris.com (D.C.); 3Oporto Higher Institute of Engineering, Oporto Polytechnic School, 4249-015 Porto, Portugal; fal@isep.ipp.pt (F.P.); mcf@isep.ipp.pt (C.F.)

**Keywords:** blockchain, edge computing, protocols, Industry 5.0, sensing, 5G networks

## Abstract

“Industry 5.0” is the latest industrial revolution. A variety of cutting-edge technologies, including artificial intelligence, the Internet of Things (IoT), and others, come together to form it. Billions of devices are connected for high-speed data transfer, especially in a 5G-enabled industrial environment for information collection and processing. Most of the issues, such as access control mechanism, time to fetch the data from different devices, and protocols used, may not be applicable in the future as these protocols are based upon a centralized mechanism. This centralized mechanism may have a single point of failure along with the computational overhead. Thus, there is a need for an efficient decentralized access control mechanism for device-to-device (D2D) communication in various industrial sectors, for example, sensors in different regions may collect and process the data for making intelligent decisions. In such an environment, reliability, security, and privacy are major concerns as most of the solutions are based upon a centralized control mechanism. To mitigate the aforementioned issues, this paper provides the opportunities for and highlights some of the most impressive initiatives that help to curve the future. This new era will bring about significant changes in the way businesses operate, allowing them to become more cost-effective, more efficient, and produce higher-quality goods and services. As sensors are getting more accurate, cheaper, and have lower time responses, 5G networks are being integrated, and more industrial equipment and machinery are becoming available; hence, various sectors, including the manufacturing sector, are going through a significant period of transition right now. Additionally, the emergence of the cloud enables modern production models that use the cloud (both internal and external services), networks, and systems to leverage the cloud’s low cost, scalability, increased computational power, real-time communication, and data transfer capabilities to create much smarter and more autonomous systems. We discuss the ways in which decentralized networks that make use of protocols help to achieve decentralization and how network meshes can grow to make things more secure, reliable, and cohere with these technologies, which are not going away anytime soon. We emphasize the significance of new design in regard to cybersecurity, data integrity, and storage by using straightforward examples that have the potential to lead to the excellence of distributed systems. This groundbreaking paper delves deep into the world of industrial automation and explores the possibilities to adopt blockchain for developing solutions for smart cities, smart homes, healthcare, smart agriculture, autonomous vehicles, and supply chain management within Industry 5.0. With an in-depth examination of various consensus mechanisms, readers gain a comprehensive understanding of the latest developments in this field. The paper also explores the current issues and challenges associated with blockchain adaptation for industrial automation and provides a thorough comparison of the available consensus, enabling end customers to select the most suitable one based on its unique advantages. Case studies highlight how to enable the adoption of blockchain in Industry 5.0 solutions effectively and efficiently, offering valuable insights into the potential challenges that lie ahead, particularly for smart industrial applications.

## 1. Introduction

In the rapidly evolving technological landscape, the combination of 5G, Industrial Internet of Things (IIoT), and advanced sensor technologies has generated tremendous interest and potential for unprecedented growth in blockchain and edge computing research. 5G, with unparalleled speed, extremely low latency, and massive connectivity, promises to revolutionize data transmission and enable seamless real-time communication between devices, paving the way for functional IIoT networks efficiently and effectively. These networks are built with a combination of smart devices and sensors, generating unprecedented amounts of data from multiple sources. Known for its secure, transparent, and decentralized nature, blockchain technology is well suited to address the challenges posed by handling and securing this huge data stream. Furthermore, 5G, IIoT, and sensors together create a dynamic edge computing environment. The combination of these groundbreaking technologies has tremendous potential to redefine businesses, improve data management, and facilitate innovation across many industries, informing that their research is a major focus of researchers in blockchain and sophisticated computing.

When discussing 5G networks, sensing is becoming an increasingly important touchstone. These devices, which include sensors that measure or detect physical phenomena and transducers, produce a large amount of data that can have a significant impact on the management of production as well as the efficiency with which it operates. The growth of sensors in the Industrial Internet of Things (IIoT) is important for the growth of automated manufacturing [1]. The vast majority of the work that humans were once responsible for in every industrial physical process can now be performed by sensors, which are more accurate and less expensive (Figure 1).

The data from 2019 are presented in the previous chart because they are the most accurate stats that are available since the pandemic crisis. Even though the use of sensors has increased over the past few years, they still present new challenges in terms of data and communication. These challenges include the necessity of ensuring that the data are accurate, the volume and the amount of time it requires to send them, and the manner in which they need to be processed.

It is possible for a variety of issues to arise if data integrity is not maintained for a variety of reasons. Some of these issues include data breaches, the loss of access to data sets, incoherence in production databases, and many others. In a recent study by IBM [3], the annual global average cost of a data breach was estimated to be USD 4.24 million, while the cost of lost business was estimated to be USD 1.59 million, or 38% of the total cost. Lost business includes business disruption and revenue losses from a system downtime, the cost of losing customers and gaining new ones, reputation losses, and decreased goodwill. Secondly, the detection and escalation costs represent USD 1.24 million (29%) of the total cost, followed by post-breach responses, costs of USD 1.14 million (27%), and notifications of USD 0.27 million (6%) (Figure 2).

In report [3], it is predicted that the amount of data that will be created in the year 2021 is estimated to be 79 zettabytes [4], even though “less than 0.5% of all data is ever analyzed and used” [5]. This occurs for a variety of reasons, including limited budgets for data analysis, poor data integration tools, manual data entry and collection processes, multiple standalone and distributed analytical tools, poor auditing procedures, and a lack of reliance on and training in system solutions to fix these issues. To address these issues, it is necessary to rely on and receive training in system solutions. 

It is necessary to have extremely low latencies of 0.5–1 millisecond [6] to solve the problems described above and to collect data at the same time. As the latency in 4G networks is approximately 200 milliseconds, only 5G networks are currently capable of providing solutions for this range. A 5G network can connect up to one million devices per kilometer [7], and it has been demonstrated that a “5G frame structure” enables devices to share available 5G bandwidth even better through a combination of time and frequency division multiplexing [8] and by mitigating signal and radio interferences. 5G networks can achieve speeds of up to 20 Mbps. One of the issues that currently exists with 5G is that there is a lack of industrial 5G equipment. However, over time, this issue will most likely be resolved within the next six months to one year. To satisfy this demand, manufacturers are not developing nor producing a new large-scale piece of hardware.

However, to overcome these obstacles, constraints, and issues, a distributed network infrastructure is required. This infrastructure must be able to deal with the outcomes of data integrity, coherence, and immutability. This is the reason why blockchain is thriving across industry. In a peer-to-peer network, blockchain operates as a shared, immutable ledger that not only records transactions but also keeps track of ownership and assets. Blockchain is a decentralized and distributed network. When a block is finished, the ledger entry for it can no longer be changed in any way.

In this document, we aimed to present the technology “assets” (such as blockchain, edge computing, sensors, and transfer data communications) needed to power industry challenges, regarding rapidness, security, and data integrity and cohesion. In the following sections, we present an architecture that gathers the technologies that should been implemented in this field.

## 2. Cybersecurity Current Overview

Digital system and infrastructure security, policies, and strategies have never been more important than now. Companies and governmental entities will spend USD 10.5 trillion yearly by 2025, which is USD 3 trillion more than in 2015 [9], with global cybercrime costs estimated to grow by 15% per year over the next five years. Also, the global spending on cybersecurity products and services will be USD 1.75 trillion, cumulatively, for the five-year period from 2021 to 2025 [10]. 

Blockchain’s main feature is a distributed ledger, which makes it more secure than both traditional distributed and centralized systems. This makes it harder for cybercriminals to act. This approach mitigates vulnerabilities by having a robust architecture (blocks) that records ownerships and transactions, ensuring integrity and data consistency. For instance, it reduces the chance of a single point of failure and makes it harder for hackers to break into a network. Similarly, once a block is added to the blockchain, it cannot be altered or deleted. This helps to ensure that the data stored on the blockchain is tamper-proof and can be trusted to be accurate. As the ledger is spread across every node of the network, it increases the complexity and difficulty for hackers to compromise, steal, or delete data. It uses cryptography to secure the data stored on the network, which helps protect against unauthorized access and ensures that only authorized parties can view or modify the data. Encryption, either symmetric or asymmetric cryptography (or cryptographic hashes), is built into the blockchain. This makes the architecture more robust. With the system’s public keys and digital signatures, it can protect any kind of edge device. Transparency, meaning that all transactions can be viewed by anyone, makes it more difficult for malicious actors to hide their activities and makes it easier for network participants to detect suspicious activities. Self-executing contracts help to reduce fraud and errors. Scaling network security with consensus algorithms makes it hard for someone to take over the network. Also, the consensus algorithm prevents anomalies without the need for a centralized and hierarchical system. A recent study from Palo Alto Networks [11] says that (a) 98% of all IoT device traffic is not encrypted; (b) 51% of threats to healthcare organizations involve imaging devices; (c) 72% of healthcare VLANs mix IoT and IT assets; and 57% of IoT devices are vulnerable to medium- or high-severity attacks and 41% of attacks take advantage of device vulnerabilities, making them an easy and desirable target for attackers. Blockchain technology eliminates the possibility of any form of distributed denial of service attack (DDoS) by not having a central point of access and by not being centralized.

## 3. Blockchains: Public and Private

There are public blockchains and private blockchains, and the distinction between the two is based on the amount of transparency that is provided. A public blockchain does not restrict who can participate, allows transactions to be verified in a way that is both transparent and decentralized, and is open to anyone who wants to use it. Bitcoin, Ethereum, and other cryptocurrencies come to mind as examples. On the other hand, a private blockchain is permissioned, which means that only participants who have been pre-approved can access it. When compared to public blockchains, these private blockchains are easier to scale and provide greater levels of privacy; however, they are also less open and centralized. Organizations frequently make use of them for the purpose of maintaining their internal records and keeping tabs on their assets. Examples include Hyperledger, etc. The fifth industrial revolution makes the most of both blockchain types. Public blockchains provide a high degree of transparency while also being decentralized. This facilitates trust, security, and accountability among users. On the other hand, private blockchains provide an increased level of control in addition to increased levels of privacy, which makes them suitable for circumstances in which sensitive information needs to be protected.

Both types of blockchain need consensus mechanisms to make sure that the network is stable and safe. Consensus mechanisms are the ideas, protocols, and incentives that make it possible for a group of nodes in different places to agree on the state of a blockchain. According to the Oxford Dictionary [12], “consensus” refers to a general agreement. There are different algorithms that are used to achieve a consensus on the blockchain. Public blockchains use decentralized consensus mechanisms, in which several nodes compete to validate transactions. Regarding Ethereum, “blockchain” means that at least 66% of the nodes on the network agree on the global state of the network [13]. The fact that every node depends on the blockchain network is shown by the fact that everyone agrees on protocols, incentives, and ideas. On the other hand, private blockchains can use either centralized or decentralized consensus mechanisms, depending on the network’s needs and goals. Most of the time, centralized consensus mechanisms are faster and more efficient but are less safe and easier to manipulate. Decentralized consensus mechanisms provide more security but are slower and less efficient.

The following subsection describes the various consensus mechanisms available for public and private blockchains.

### 3.1. Proof of Work (PoW)

Proof of Work (PoW) is a consensus algorithm used to secure blockchain networks and validate transactions. PoW is the consensus mechanism used by Bitcoin, Litecoin, and Dogecoin. It was created by Satoshi Sakamoto, the creator of the Bitcoin blockchain. Ethereum, until Ethereum 2.0, uses PoW. The idea behind PoW is that the puzzle is difficult to solve but easy to verify. Proof of Work involves scanning for a value that, when hashed, such as with SHA-256, begins with a number of zero bits. The average amount of work needed grows by a factor of the number of zero bits needed, and this can be checked by running a single hash. For our timestamp network, we perform Proof of Work by increasing a nonce in the block until we find a value that provides the block’s hash with the required zero bits [14]. When the block meets the hash requirement of zeros, the block is “chained” to the network and can no longer be edited. A consensus convinces attackers that if they try to change a block, they will have to redo Proof of Work for the existing block in the chain of blocks and, if they do that, they will have to redo other blocks as well. The system provides users with bitcoins, whose value has been going down over time, when they find a mistake. Thus, centralization could happen if some users are rewarded more than others, which would favor the computing power of the rewarded users. Moreover, centralization may compromise data integrity. PoW is the most prevalent consensus mechanism for modeling public blockchains in general. Proof of Work (PoW) is not usually used in private blockchains because it was made for public blockchains, which need a decentralized consensus and protection from bad actors. Ethereum Classic is a private version of Ethereum that uses a PoW consensus.

PoW has been an important part of the blockchain industry since the beginning, but it has not contributed much for Industry 5.0. PoW is hard to use for integrating advanced technologies because it uses a lot of energy, is hard to scale up, and could become centralized.

### 3.2. Proof of Stake (PoS)

Proof of Stake (PoS) is an alternative to Proof of Work (PoW) that was implemented to address the shortcomings of PoW. PoS is better for the environment and uses less energy than PoW because it does not require miners to solve hard math problems to verify transactions. In PoS, the validation of transactions is carried out by validators, who are selected based on the amount of cryptocurrency they hold in the network. This is why PoS is also referred to as “staking” because, as they have a stake in the network, the validators have a reason to check transactions honestly. If they check transactions with bad intentions or if the network is attacked, they could lose the tokens they staked. The process starts with a proposer, then a proposed block, and finally the validation of the proposed block [15]. The amount of cryptocurrency a validator has in the network affects how likely it is that they will be chosen as a validator. Bigger token ownerships have more chances to be selected, even though the selection is random. Since there is no reward for mining, it encourages more nodes to take part in creating and validating blocks, which saves energy. In PoW, the validation process becomes more difficult as the network grows, which slows down the process of validating transactions. In PoS, the validation process remains constant, regardless of the size of the network. This makes it a more scalable solution, especially for blockchain networks with many users. The PoS is not without its drawbacks. One of the main criticisms of PoS is that it is vulnerable to the “nothing at stake” problem. In this case, validators have no reason not to switch to a different chain if a new chain with better rewards is made. This can lead to a situation where the original blockchain becomes vulnerable to attacks. Despite being a “fairer” mechanism, it still has flaws for individuals with lower holdings because ownership is correlated with the likelihood of selection, increasing the potential for centralization. Again, the mechanism may compromise decentralization, having in mind that smaller networks are less efficient at staking, leading the network for centralized nodes. PoS is used in both public and private blockchains. Ethereum (as of Ethereum 2.0), Cosmos, Tezos, Algorand, EOS, etc. are some public blockchains that make use of this consensus. Hyperledger Besu, Corda, Quorum, Chain Core, etc. are examples of private blockchains.

Proof of Stake has several advantages that make it well-suited for contributing to Industry 5.0, including its energy efficiency, scalability, and decentralization.

### 3.3. Delegated Proof of Stake (DPoS) 

DPoS relies on a select group of individuals, known as “delegates” or “witnesses”, who are responsible for achieving a consensus during the block validation [16] to validate transactions. These delegates are elected by the community, and they are incentivized to act in the best interest of the network by being rewarded with transaction fees. The block validation is subject to a voting system by stakeholders to choose the external validator. The key difference between DPoS and other consensus mechanisms is that DPoS uses a democratic voting process to select its delegates. This process allows for a more efficient and secure network, as it eliminates the need for large amounts of computational power to validate transactions. One of the major benefits of DPoS is its ability to process transactions at a much faster rate than other consensus mechanisms. This is because DPoS networks have a much smaller number of delegates who are responsible for validating transactions, which allows for a more streamlined process. DPoS networks are also more scalable than other consensus mechanisms. This is because the number of delegates can be adjusted to meet the demands of the network, which means that the network can continue to grow and process more transactions without sacrificing its speed or security. This mechanism promotes energy efficiency, but it may lead to further centralization and questionable ethical behavior because blockchain-based validation can stifle the spread of public opinion due to the limited number of delegates. Nonetheless, cases may force the mechanism to rely on centralized processes because a small number of token holders can impact and influence networks. DPoS is used in both public and private blockchains. EOS, Ark, TRON, and BitShares are some examples of public blockchains that use this consensus. Hyperledger Iroha, Symbiont, and Eris Industries are some examples of private blockchains that use this consensus.

Delegated Proof of Stake has several advantages that make it well-suited for contributing to Industry 5.0, including its fast block times (speed), low latency, efficiency, scalability, and decentralization.

### 3.4. Byzantine Fault Tolerance Family (BFT)

The BFT is the feature of a distributed network to reach a consensus (agreement on the same value), even when some of the nodes in the network fail to respond or respond with incorrect information [17]. It aims to protect the system against failures by employing collective decision making on correct and incorrect nodes—this enables a reduction in faulty nodes. The concept is inspired by the well-known Byzantine Generals’ Problem [18]. The problem represents several Byzantine divisions, each led by a general and stationed outside an enemy city where generals can communicate via messages. Before they take any action, they must agree on a common strategy. However, some generals are not trustworthy, and they will try to avoid loyal generals to reach an agreement. Facing this problem, generals must decide what to do, based on a strong majority of generals to have a common attack plan at the same time. The generals must have an algorithm to ensure that loyal generals decide the same action against the minority of the not trustworthy generals’ bad plan. Finally, untrustworthy generals may carry out the bad plan without causing any harm because the majority of loyal generals carry out the same strategy at the same time, meeting a reasonable plan and agreement.

In conclusion, an agreement problem can be solved if most n processors are faulty, which means that strictly more than two-thirds of the total number of processors should be honest (if we have 3n + 1 working processors), allowing tolerance for n faults [18].

The consensus mechanisms are based on the BFT concept and different approaches, although they all rely on three key properties: (1) safety, (2) liveness, and (3) fault tolerance. (1) A consensus protocol is determined to be safe if all nodes in the network agree on the same state of the blockchain. This means that the network will always reach a consensus and all nodes will have the same view of the blockchain, even in the presence of network partitions or other failures. This is also referred to as the “consistency” of the shared state. (2) A consensus protocol guarantees liveness if all non-faulty nodes participating in a consensus eventually produce a value, meaning the network can continue to operate even in the presence of failures or other issues. (3) A consensus protocol provides fault tolerance if it can recover from the failure of a node participating in a consensus. This is performed with the help of redundancy, finding and fixing errors, and other methods [19]. BFT algorithms are designed to ensure that the network can reach a consensus even in the presence of failures, and that the network continues to operate even in the presence of network partitions or other issues. The BFT concept is widely used in distribution systems.

BFT has some limitations, such as (1) scalability, (2) latency, and (3) resource consumption. (1) BFT algorithms need a lot of messages to be sent between network nodes, which can make it harder for the network to grow as the number of nodes increases. (2) BFT algorithms can also be slow, with high latency between nodes in the network. This can result in slow transaction processing times and a less responsive network. (3) BFT algorithms can use a lot of resources, such as a lot of processing power, memory, and network bandwidth. This can limit the deployment of BFT algorithms in resource-constrained environments.

To overcome BFT limitations, the PBFT algorithm was developed. PBFT is a modification of BFT that aims to make a consensus in a blockchain network more scalable, efficient, and useful. PBFT uses a pre-consensus protocol to reduce the number of messages sent between network nodes. This helps to improve scalability and reduce latency. Also, PBFT uses a more efficient way to reach a consensus. This makes the network use less resources and lets it work in places where resources are limited. PBFT assumes that nodes may act maliciously and tries to overcome this by using a majority agreement mechanism, where the majority of nodes must agree on a single value. This helps to ensure that the network reaches a consensus on a correct value, even in the presence of malicious nodes [20]. PBFT relies upon a combination of digital signatures, cryptographic hash functions, and a majority agreement to ensure the authenticity and integrity of the data exchanged between nodes and to reach a consensus on a correct value [21].

Like pBFT, the BFT comprises multiple other implementations, such as (1) iBFT, (2) dBFT, (3) Tendermint, (4) mBFT, (5) fBFT, (6) DiemBFT, etc. 

(1) Istanbul BFT [22], also known as iBFT, is intended for use in large networks containing thousands of nodes. It uses a committee-based approach to reach a consensus, where a group of nodes are selected to reach a consensus. It is used in private blockchain networks. For example, iBFT is used with Quorum, Pantheon, etc. 

(2) Delegated Byzantine Fault Tolerance, or dBFT [23], is designed to handle malicious nodes in the network and ensures the authenticity and integrity of the data being exchanged between nodes. dBFT is a combination of the Delegated Proof of Stake (DPoS) and Byzantine Fault Tolerance (BFT) algorithms, which means it offers scalability and high performance, such as DPoS, and the security and reliability of BFT. In dBFT, token holders vote for a set of nodes to act as bookkeepers, who then reach a consensus on the next block in the chain using a BFT consensus mechanism. When compared to other BFT consensus algorithms, dBFT is known for being fast and having low latency. It is used in public blockchain networks, for example, in NEO. 

(3) Tendermint [24] uses a leader-based approach to reach a consensus, where a main node acts as the leader and is followed by backup nodes. Tendermint is known for its fast finality and low latency compared to other BFT consensus algorithms. When compared to other BFT consensus algorithms, Tendermint is known for being fast and having low latency. It is used in both public and private blockchain networks. For example, the Cosmos network, a decentralized public network of independent blockchains, and Binance Smart Chain, a decentralized private network. 

(4) Modified Byzantine Fault Tolerance, or mBFT [24], is also designed to improve its performance and scalability. mBFT uses a combination of digital signatures, cryptographic hash functions, and message broadcasts to reach a consensus on the next block in the chain. Unlike traditional BFT algorithms, mBFT uses a simpler communication protocol and relies on a smaller number of nodes to reach a consensus, making it more efficient and scalable. Despite its improved performance, mBFT still maintains the security and reliability of traditional BFT algorithms. It is used in private blockchain networks, for example, in Hyperledger Besu. 

(5) Fast Byzantine Fault Tolerance, or fBFT [25], is optimized for fast block confirmation times and high transaction throughput, making it ideal for decentralized systems that require quick, reliable transaction processing. It is used in private blockchain networks, for example, in the Chain network. 

(6) DiemBFT [26] is directly based on PBFT and indirectly based on BFT and aims to provide a stable, secure, and scalable platform for digital transactions and financial applications. It is based on the HotStuff (https://arxiv.org/pdf/1803.05069.pdf (accessed on 14 August 2023)) protocol, built on pBFT, and aims to increase efficiency by reducing the number of messages and, thus, communications between nodes, while maintaining pBFT security and efficiency. The leader has a bigger role, rather than interacting between nodes. Security is improved because a leader is selected randomly (the leader/follower concept). Members are assumed to be nodes, and transactions are sent to them by clients, operating through a shared mempool [27]. A rotation rule allows nodes to become leaders, based on HotStuff, with controlled timeouts, and can propose new blocks to be added to the chain that must be approved by followers. When a block obtains the majority of votes, it is added to the chain and obtains a “quorum certificate”, which is spread across the network for validation. If the process goes well, it is stored in the chain. This allows the consensus to be faster at a minimum cost: compared with 7 transactions per second, the DiemBFT consensus allows for 1000 transactions per second. Although speed and efficiency features exist, there are some problems with security, integrity, and privacy. Nevertheless, there is a centralized consensus. It requires power to compute because nodes must commit at least USD 10 million in Diem stable coins to participate. This consensus is used with Diem and Facebook’s Libra. It is used in private blockchain networks, for example, in Diem Blockchain.

By comparing iBFT with PBFT we know that iBFT is more scalable than PBFT and has a faster finality compared to PBFT. Instead of “leaders” and backup nodes, iBFT uses “proposers” (who act as leaders) and “validators” (who act like backup nodes)—they can validate blocks but have no active role on the consensus protocol. In each round, validators may choose a new proposer responsible for adding the next block for validation. The biggest difference between pBFT and iBFT is that validators in iBFT can change, while in pBFT they are static, which means, by principle, that validators in iBFT are more truthful and involved. Like other consensus mechanisms, pBFT and iBFT operate where malicious nodes do not exceed 66% of all nodes and block states do not require confirmation, similarly to how they operate on public consensus mechanisms (trust is assumed in private mechanisms). Also, compared with the public consensus, pBFT and iBFT consume less energy, considering the inexistence of minors to solve mathematical operations. As they use a large number of messages to keep track between blocks and collective decisions, these consensus systems work better with a limited number of nodes. More nodes imply that more actions need to be performed. Even though pBFT and iBFT are still vulnerable to attacks and compromise security, a node may be attacked, and the leader may manipulate other nodes. Due to the leader/proposer, a consensus allows nodes to be controlled in a closed system, which is different from public consensus mechanisms (where nodes are open and free).

### 3.5. Distributed Proof of Security (dPoSec)

The dPoSec consensus algorithm was developed with the goal of simplifying the operation of the blockchain network during communication and increasing the amount of work that can be performed in a set amount of time [17]. It intends to function in a highly secure mode while also scaling itself towards a Phase-3 solution, which will make it a significant improvement in the field of blockchain consensus algorithms.

In order to provide a blockchain solution that is both quick and scalable, the dPoSec algorithm combines the most beneficial aspects of the Proof of Stake (PoS) protocol and the Byzantine Fault Tolerance (BFT) protocol. It incorporates advanced security measures, such as trust establishment, among nodes to ensure a fast, scalable, and secure network operation. It is a significant improvement over traditional consensus algorithms and represents a significant step forward in the evolution of consensus algorithms.

As the dPoSec is built on the Ethereum Virtual Machine (EVM), it is able to execute all smart contracts, and offers a flexible platform for the development of decentralized applications. This makes it possible for developers to build a diverse set of decentralized applications on the platform. Due to this, it is a flexible solution that can be implemented in a variety of fields, including the healthcare industry, the financial sector, and others.

dPoSec is a balanced protocol that enables versatile platforms because, in addition to its robust security features, it also provides on-demand privacy and efficient peer-to-peer discovery. This makes it a better solution for any situation. This algorithm was developed to be highly scalable, and its architecture makes use of ultimate blockchain primitives to improve both its performance and its efficiency. dPoSec’s blockchain primitives are designed to be more efficient than those of other blockchains, which results in a number of benefits, including a reduction in the amount of time needed to process transactions and an increase in scalability. As dPoSec is an on-demand privacy platform, users are able to protect the confidentiality of their data and take advantage of the privacy benefits that come along with using the service. This means that users can choose which information they want to share and with whom, providing them with full control over their data and giving them the ability to make decisions based on all gathered data. The peer-to-peer discovery feature of the platform makes it easy for nodes to find each other and establish connections, thereby lowering the network’s latency and increasing its overall efficiency. This is achieved through incentivized schema, strong consistency across nodes and across verge clusters (decentralized mesh of nodes across shard-like structures known as verge clusters).

One of the most significant obstacles that blockchain networks must overcome is the requirement for a high level of coordination and communication between nodes. This can make the network move more slowly and raise the likelihood of it failing altogether. dPoSec is able to circumvent this difficulty by employing a unique validator selection process. This method was developed with the intention of lowering the likelihood that malicious actors will compromise the network and raising the level of network security overall. In order to maintain the reliability and safety of the network, validators are chosen using a variety of criteria, including the amount of stake they hold, their reputation, and their performance.

dPoSec makes use of a sophisticated reward mechanism for validators in order to ensure that the network’s integrity is preserved and to thwart any attempts by malicious actors to compromise it. This incentivizes good behavior and punishes malicious actors, helping to maintain the network’s security. The algorithm also eliminates the danger of “nothing at stake” attacks, which occur when validators carry out malicious behavior without fear of any repercussions as a result of their actions. dPoSec addresses this issue through its punishment mechanisms, which penalize malicious validators and incentivize good behavior.

dPoSec is a secure and reliable consensus mechanism because it uses advanced security measures to increase trust in the running nodes. These measures include cryptographic signatures and consensus algorithms, among other things. This ensures that the network is protected against a variety of attacks, including those that exploit vulnerabilities in the consensus algorithm. This also protects the network from being compromised. For example, the use of cryptographic signatures helps to ensure that the network’s integrity is maintained, even in the event that one or more of its nodes are breached.

dPoSec was developed with a particular emphasis on security, efficiency, and scalability in order to cater to the requirements of a wide range of sectors and assist those sectors in maximizing the potential of blockchain technology. This algorithm is highly adaptable and can be tailored to the particular requirements of each sector, which enables it to function as a solution that is both versatile and scalable. For instance, it can be used in the financial sector to ensure the safety and efficacy of payment transactions, and it can also be used in the healthcare sector to secure and manage electronic medical records. Both of these applications can be found in the financial sector. It is designed to meet the needs of a variety of industries and to assist those industries in making use of the benefits offered by blockchain technology. It accomplishes this by combining advanced security measures with a focus on scalability [17].

Both public and private blockchains are compatible with and capable of using dPoSec. This consensus is used by the Naoris Protocol, which is a hybrid public/private blockchain as per the end need.

dPoSec may contribute considerably to Industry 5.0 by offering a reliable and effective consensus mechanism for industrial blockchain applications. For instance, it can be used to safeguard the accuracy of information gathered from Internet of Things devices in a supply chain management system or to safely track the flow of goods through the supply chain.

### 3.6. Proof of Elapsed Time (PoET)

PoET is designed to address the problem of energy waste in Proof of Work (PoW) and Proof of Stake (PoS) systems. The main goal of this consensus is to reduce energy consumption. Each node in the network waits for a random amount of time before it can participate in block validation. This waiting time is verified by a trusted entity, called the “Validator”, who verifies that the node indeed waited for the specified amount of time. Developed by Intel, this consensus enables a tool to solve the problem of randomly selecting a leader who makes decisions about mining permissions and to block winners, aiming to spread winnings across a large number of participants [28]. By managing waiting times and managing energy consumption, the shortest randomly generated waiting time by a node wins the block. This consensus is fairer because it promotes a way of rewarding centralization and random leader selection; thus, this consensus remains probabilistic instead of deterministic. Hyperledger Sawtooth uses this consensus.

PoET can be used in both public and private blockchain networks. Hyperledger Sawtooth is a public blockchain that uses this consensus. Intel SGX is a private blockchain that uses this consensus.

### 3.7. Proof of X (PoX)

Revolutionary change is often accompanied by growing pains, and the world of blockchain technology is no exception. As the field continues to evolve, new problems arise and demand new solutions. One such area of innovation is the development of many such consensus algorithms, and discussing them all will be out of the scope for this paper. There is currently more than 30+ different consensus algorithms that support either public or private blockchain networks, or both, and the list is constantly growing [29]. Some of the other most popular Proof of X algorithms include the following:
Proof of Capacity/Space○This consensus algorithm uses disk space instead of computational power to validate transactions. The idea is that disk space is cheap and readily available, making it a more sustainable and energy-efficient alternative to PoW.
Proof of Burn○This consensus algorithm requires users to “burn” tokens by sending them to a public address with no known private key. This reduces the supply of tokens in circulation, making it more difficult for attackers to accumulate a large amount of tokens to launch an attack.Hybrid Models○As the name suggests, hybrid models combine two or more consensus algorithms to take advantage of their strengths and mitigate their weaknesses. For example, a hybrid of PoW and Proof of Stake (PoS) might use PoW to secure the network against attacks, and might use PoS to validate transactions.Directed Acyclic Graph (DAG)○A DAG is a type of data structure that can be used to create a distributed ledger. In a DAG-based blockchain, transactions are validated based on their position in the graph, rather than through a traditional consensus mechanism.


By constantly pushing the boundaries and exploring new solutions, the field of blockchain transaction’s reaches its finality if it the following aspects occur: uses probabilistic or deterministic algorithms; if participants have or do not have permission to join the network and contribute to its maintenance [30]; uses tokens needed to operate; exceeds the scalability capacity in order to grow the network; transaction speeds are classified; computes any costs of participation in the infrastructure, computing power, and operations in or on the network; does not reach the required energy efficiency; the network does not trust the processes and transaction independently from participants and entities; is not resilient to maintain a stable and common state across blocks; if decentralization is not well implemented; tolerance is allowed in order to make decisions and validate blocks; if decentralization, security, and scalability are addressed, at least theoretically, known as blockchain trilemma; the layer type of blockchain, type 1, base layer, type 2, blockchains built in base layers type 1 and type 3, and decentralized blockchains and protocols are not used.

### 3.8. Discussion

As expected, there is no “magic” consensus for all scenarios at sight, services, or industries. Issues, such as the size of the infrastructure, the expected time of interactions, the classification of participants, the data to be exchanged, and so on, must be addressed. Nevertheless, there are some considerations to keep in mind, such as energy efficiency, decentralization, security of actions and transactions, the expected growth of devices, always keeping some guidelines in mind for achievement, business success, and costs. Trust, resilience, and tolerance are properties that should help in a decision concerning which consensus to adopt. This should also be planned according to the technologies that are available to the organizations and infrastructures where the consensus mechanism is the very heart of every blockchain solution [31]. Ultimately, any choice about the adoption of a blockchain solution should be made with a focus on the consensus process that underpins the solution, and with a comprehensive grasp of the technologies and infrastructures that are currently accessible.

## 4. Industry 5.0 and Edge Computing

The digitalization of industries is an ongoing process, and is also known as Industry 5.0. Since automation, this is probably the most important stage of all time, with the aim for better management and optimization of all aspects of the manufacturing processes and supply chains. Like Harald Lesch said, digitalization is the purest form for monetizing time [32]—to achieve this goal, there are a set of technologies and systems that are needed, such as sensors, 5G networks, and (at the core) robust distributed systems, to gather and process all the data. As presented previously, the growth of sensors for different purposes is exponentially growing, while prices are continuously dropping. However, to make use of all end-of-line devices, computers are needed to gather and store all the data they transmit (1), process all the data in a way for achieving knowledge, validation, and refinement of the decisions in the supply chain (2) and, nevertheless, do it in the smallest time frame window that can be achieved (3) with the smallest amount of energy needed.

Companies such as Bosch [33], Airbus [34], or BMW [35] have now implemented 5G networks in their production lines with the aim of gathering data to improve efficiency. Along with secure and reliable communication, another important aspect is to achieve high-speed communication.

To obtain results from 5G, the use of remote servers in an edge computing architecture can provide greater power flexibility and overall process performance, resulting in lower implementation costs and greater sustainability. In an edge computing architecture, kilometers of cables are not required to connect sensors to communicate with control systems [33]. Let us proceed to a briefing on edge computing enhancements.

### Edge Computing Enhancement for Industry

Edge computing is a distributed computing paradigm that allows data processing to occur closer to the source of data, rather than in a centralized location. The main goal of edge computing is to reduce the amount of data that needs to be sent to centralized data centers, which can improve the speed and reliability of data processing, as well as reduce the costs associated with data transfer and storage.

It is expected that the edge computing market size is projected to reach USD 101.3 billion by 2027, at a CAGR of 17.8% [36], from components to applications. In a comparison study between cloud computing and edge computing [37], wireless networks are more suitable for sensor communications than wired networks for the cloud; thus, more mobile implementation is desirable as well as greater scalability, allowing a wider distribution of digital devices. Due to the reduction in cables, less power consumption is needed, reducing operational costs as well as storage costs. Other advantages are related to privacy and security, reducing data leakage during transmissions; low latency rates (thanks to a decentralized network), enabling better data transmission and real-time results; more manageable data for analytics; and better interoperability (thanks to the computing processing moving to the edge), which eliminates the need for universal device standards that are not yet defined.

Also, industrial process monitoring and predictive maintenance are two of the most common use cases for edge computing implementations. Efficient communication across highly complex SCADA (supervisory control and data acquisition) systems to manage the high volumes of data from sensors and PLCs (programmable logic controllers) and the ability to track a variety of metrics and monitor the performance of machinery is becoming a standard in state-of-the-art factory plants [38].

## 5. The “New Industry” Challenges

The new industrial revolution, also known as Industry 5.0, has significantly improved technology and changed how various industries function. Despite all of its advantages, Industry 5.0 has also created important challenges that must be overcome for its successful adoption. Standardization, which is essential for seamless communication and collaboration across many platforms and systems, is one of the main issues. Latency, or the delay in data transport, is another important restriction that presents a problem for real-time applications. Additionally, to guarantee the secure and effective functioning of Industry 5.0 technologies, security, scalability, and dependability are significant challenges that must be addressed. For Industry 5.0 to reach its full potential and for industries all around the world to experience sustainable growth, it is essential to address these obstacles.

## 6. Major Challenges in Blockchain Adoption

This section presents, in the form of an itemized list, the concerns regarding the implementation of blockchain architectures in industries and the consensus mechanism characteristics (Table 1).

### 6.1. Devices Incompatibility

The adoption of blockchain in low-end IoT devices is challenging due to their limited processing power, memory, and energy resources. Low-end IoT devices may struggle to perform complex blockchain computations, resulting in slow transaction processing times and overall system performance degradation. These limitations make it difficult to adopt blockchain as a direct solution for Industry 5.0 legacy-based systems that rely on low-end IoT devices. Therefore, careful consideration must be provided to the performance requirements of blockchain solutions and the capabilities of low-end IoT devices when implementing blockchain in Industry 5.0 applications.

### 6.2. Scalability

The massive amount of data produced by numerous linked devices presents a significant challenge for blockchain technology. Blockchain is known for being slow and not scalable, potentially causing a bottleneck in the adoption of devices in Industry 5.0 systems. As a result, implementing blockchain in Industry 5.0 applications requires careful consideration of scalability issues to ensure efficient processing of the vast amounts of data.

### 6.3. Interoperability

The integration of Industry 5.0 devices with blockchain systems may be challenging due to the use of multiple protocols and standards that are incompatible with blockchain technology. The diverse range of protocols used in Industry 5.0 devices could potentially hinder the seamless integration of blockchain, making it difficult to establish effective communication and collaboration. Therefore, addressing interoperability issues is crucial for the successful integration of Industry 5.0 devices with blockchain technology.

### 6.4. Security

Although Industry 5.0 devices are generally considered secure, they may not be as secure as blockchain technology. The interconnectivity of devices in Industry 5.0 makes them vulnerable to security breaches, which could potentially compromise the integrity of the entire blockchain network. Therefore, ensuring the security of Industry 5.0 devices is critical to the successful implementation of blockchain technology in Industry 5.0 applications.

### 6.5. Privacy

Industry 5.0 devices often collect sensitive data, which must be protected to ensure the privacy of individuals. Integrating blockchain with IoT requires careful consideration to ensure that data privacy is not compromised. Robust security measures must be established to safeguard sensitive data from unauthorized access and ensure the integrity of the blockchain network. Therefore, careful attention must be provided to the privacy and security implications of integrating blockchain with IoT to ensure the successful implementation of Industry 5.0 applications.

### 6.6. Cost

The integration of blockchain technology into an Industry 5.0 system can be a costly affair and may require significant investments in infrastructure, software, and hardware. The deployment of blockchain technology may require additional resources to support the increased computational demands and data storage requirements of the system. Therefore, implementing blockchain in Industry 5.0 applications requires careful consideration of the costs involved to ensure the feasibility of the integration.

### 6.7. Trustlessness

While trustlessness is a key benefit of blockchain technology, it also presents a significant challenge. While this eliminates the need for trust in centralized entities, it can create vulnerabilities in the system that malicious actors can exploit. One of the significant security challenges presented by trustlessness is the potential for 51% attacks. A 51% attack occurs when an entity gains control of over 50% of the computing power of the blockchain network. Furthermore, the absence of trusted intermediaries in a trustless system can also make it challenging to identify and address security breaches, as there is no central authority to oversee the transactions.

### 6.8. Storage

Another significant challenge in adopting blockchain for devices is the issue of storage space. Industry 5.0 devices generate massive amounts of data, and recording all the data on the blockchain network can quickly exhaust the storage capacity of the devices. Moreover, storing large amounts of data on the blockchain can increase the overall size of the blockchain, making it less scalable and less efficient.

## 7. Gathering Blockchain and Edge Computing: A Proposal Distributed Computing Solution

While many blockchain solutions have been proposed to address various challenges, none of them resolve all the issues. As a result, the adoption of blockchain has been challenging, as individual blockchain solutions may not be sufficient to address all the problems. As a result, a set of components, devices, applications, architectures, and technologies may address the challenges and needs of the industry. As an example, Bosch Security Systems, S.A, located in the northern region of Portugal, is now using more than 5000 5G sensor devices to improve the production on their products. This industry is using more sensors each day to collect data and process data to improve maintenance (predicted), reduce waiting times, and produce more with higher cost-efficient ratios. Despite the implementation of these changes, most of the processing is performed with an edge computing strategy—which means that it utilizes local processing, which is good for data privacy and rapid data analysis—but cannot be integrated into a blockchain network (Figure 3).

If the dPoSec consensus mechanism, designed with a security-first approach for industrial blockchain applications, has the potential to significantly enhance Industry 5.0 by providing improved security, scalability, and interoperability, its integration into the industrial ecosystem will enable more reliable and accurate information gathering from IoT devices across diverse domains, making it easier to implement and manage complex blockchain solutions.

As previously stated, dPoSec has been designed to address almost all the challenges discussed earlier, and can play a crucial role in Industry 5.0. It offers a customized EVM-based robust and efficient consensus mechanism designed with a security-first approach for industrial blockchain applications, and can be utilized to ensure the accuracy and reliability of information gathered from Internet of Things (IoT) devices across various domains. By utilizing dPoSec, Industry 5.0 can benefit from enhanced security, scalability, and interoperability, making it easier to implement and manage complex blockchain solutions.

If the dPoSec consensus mechanism leverages the synergy of edge computing and local processing, which play a pivotal role in bolstering the trust and security of the blockchain network, and enable IoT devices to actively participate in the consensus process through local processing, dPoSec can thus enhance network security and efficiency. Moreover, if the mechanism’s capacity can learn from network operations and develop tailored defenses for associated devices, it may foster adaptability and collaboration, instilling confidence among devices. Consequently, dPoSec has the potential to effectively address various key challenges in Industry 5.0, including scalability, interoperability, and security, thereby simplifying the implementation of sophisticated blockchain solutions.

dPoSec is a consensus mechanism that combines the power of edge computing and local processing to establish trust and enhance the security of the blockchain network. By utilizing local processing power, dPoSec can provide IoT devices with the ability to participate in the consensus process, which enhances the security and efficiency of the network. Furthermore, dPoSec is designed to learn from the knowledge gained during network operation and to develop defenses for all associated devices. This enables the network to adapt and evolve, building confidence among devices and promoting collaboration. As a result, dPoSec can help to overcome some of the most significant challenges of Industry 5.0, such as scalability, interoperability, and security, making it easier to implement complex blockchain solutions.

How can the implementation of dPoSec in Industry 5.0 improve the security of the blockchain network, allowing IoT devices to operate with bigger trust and efficiency? And, thus, what are the specific advantages of this, such as improved data accuracy, faster processing times, and increased resilience against cyber attacks, that can be attributed to the combination of edge computing and local processing provided by dPoSec, ultimately unlocking the full potential of Industry 5.0 in the realm of industrial blockchain applications?

By leveraging dPoSec, Industry 5.0 can benefit from an enhanced security posture that enables IoT devices to operate with greater trust and efficiency within the blockchain network. This can lead to improved data accuracy, faster processing times, and increased resilience against cyber attacks. Ultimately, the combination of edge computing and local processing provided by dPoSec offers a more reliable and effective consensus mechanism for industrial blockchain applications, helping to unlock the full potential of Industry 5.0.

You may have a blockchain network in a powerplant, gathering and controlling data across servers, desktops, and tablets, and you also may have data sensors in devices, gathering data for local processing in the form of edge computing. However, the two worlds do not exchange information.

Our proposal to integrate the dPoSec protocol as a blockchain network with sensor devices, providing information for the blockchain network and edge computer processing.

Figure 4 shows dPoSec blockchain networking with a plethora of connected verge clusters and their separate meshes. Each mesh may be composed of several IIoT sensors with blockchain firmware. What are the potential strategies or approaches to overcome the challenge of integrating sensors and machinery for data acquisition, which require support the upload of blockchain firmware for posterior edge processing, given the prevalence of proprietary and closed solutions offered by various manufacturers?

To address the challenges in this field, sensors and machinery, which perform data acquisition for posterior edge processing, need to be prepared to support the upload of blockchain firmware. Despite the fact that there are several manufacturers that support this technology, most of them are proprietary and closed solutions. dPoSec-based networks comprise of the following aspects (Figure 5):

## 8. Conclusions

As a result, a set of components, devices, applications, architectures, and technologies are now available to drive industry digitalization and efficiency while addressing new challenges. The touchstone at the present moment must be the integration of these built-in simple use cases and their spread in this field.

The digitalization of industries is an ongoing process, also referred to as Industry 5.0, which involves the use of various advanced technologies, such as artificial intelligence, robotics, IIoT, cloud computing, virtual reality, augmented reality, and blockchain. It is transforming how organizations operate from the stage of production to customer service, enabling the integration of data from a variety of sources and helping in the automation of processes. It is also reducing the need for human resources and increasing the efficiency of many organizations [39]. The use of digital technologies has enabled companies to optimize their operations and create new products and services that are more personalized to customers [40]. Combining the discussed technologies will enable improvements in supply chains and within the standard procedures of industries.

Sensorization can provide greater accuracy and real-time data, allowing for more informed decisions to be made in real time. Blockchain consensus can help to provide a secure, truthful, and immutable platform for data and transactions, eliminating the need to verify transactions through a centralized authority. Edge computing can reduce latency and increase data processing speeds, allowing for quicker decisions to be made and for more complex tasks to be performed. 5G can provide faster internet speeds and increased bandwidth, allowing for more data and devices to be transmitted and processed faster. Overall, the combination of these technologies can offer organizations and individuals greater efficiency, security, transparency, and cost reductions.

Also, edge computing and blockchain complement each other: they are not inherently similar. Edge computing can also optimize the performance of blockchain-based systems by reducing latency and improving response times.

Nevertheless, challenges and constraints arise for companies when facing new outcome standards. Lack of knowledge and expertise: companies may lack the knowledge and expertise to implement Industry 5.0 technologies, making it difficult to take advantage of the benefits they offer (1) [41]. High cost of implementation: implementing Industry 5.0 requires significant capital investment in expensive technologies, such as artificial intelligence, robotics, and Internet of Things (IoT) systems (2) [42]. Security risks: connecting systems and data to the Internet creates potential security risks, and companies must invest in security measures to protect their data and networks (3) [43]. Adaptability and scalability: companies must be able to quickly adapt to changing technologies and customer needs, which can be difficult for companies to do without the right infrastructure in place (4) [44]. Regulatory restrictions: depending on the industry, there may be regulatory restrictions in place that limit the use of certain technologies, such as autonomous vehicles or drones (5) [45].

One of the key challenges in implementing edge computing is ensuring the security and privacy of the data that is being processed and transmitted. Data breaches and cyberattacks are becoming more and more likely as more gadgets are connected to the internet. Blockchain can fill this need by offering a safe and unhackable method of data storage and distribution, for example, a smart factory where hundreds of sensors are gathering information on the functionality of various machines and pieces of equipment. To enable predictive maintenance and other uses, these data must be analyzed in real time, but they must also be secured to prevent unwanted access. The factory can guarantee that only authorized parties have access to the data and that any modifications or updates to the data are documented and confirmed by using blockchain to store and exchange these data.

Similarly, to enable safe and effective navigation, self-driving cars need to handle a lot of data in real time, including data from cameras, lidar, and other sensors. Cars can make judgments more quickly and react to changing situations in real time if edge computing is used to process these data locally, rather than sending them to a centralized cloud-based system. As a car is gathering information about its surroundings and the actions of other drivers, this also raises questions regarding data security and privacy. The automobile can make sure that only authorized parties have access to the data, that any modifications or updates to the data are recorded and confirmed, and that the data are securely stored and shared via blockchain.

Extending businesses surely increases effectiveness and transparency by employing sensors and other IoT devices to monitor the movements and conditions of their products, and the data are securely stored and shared by using blockchain technology.

When combined, edge computing and blockchain—two of the most exciting and promising technologies of our time—offer a potent solution that can present a wide range of new use cases and applications. Edge computing and blockchain are poised to disrupt numerous industries as well as how we live and work by enhancing data security and privacy, enabling new business models and revenue sources, and enhancing the efficiency and transparency of supply chains.

It is important to emphasize that nothing similar has been performed thus far. Only works related to the implementation of blockchain distributed systems were created, although they were never merged with edge computing processing, due to its complex application.

The scientific contribution of this work consists of the creation of an innovative architecture that combines a blockchain protocol used as firmware in industrial sensors, which collect data that will be used with edge computing processing in various industries. The use of the blockchain dPoSec protocol is the most reliable way to ensure the safety of industries and data integrity.

## Figures and Tables

**Figure 1 sensors-23-09174-f001:**
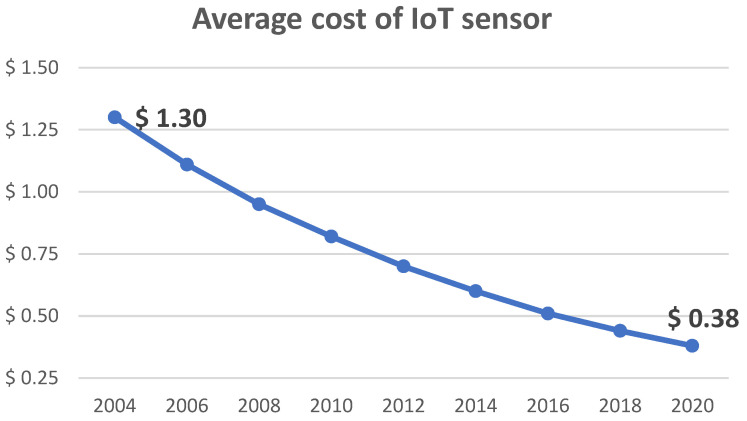
Average cost of IoT sensor [2].

**Figure 2 sensors-23-09174-f002:**
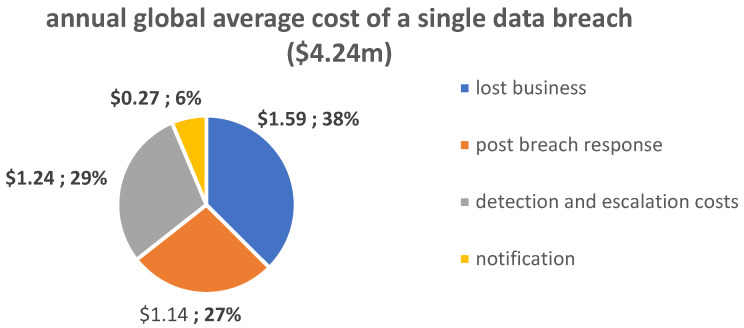
Average cost of a data breach [3].

**Figure 3 sensors-23-09174-f003:**
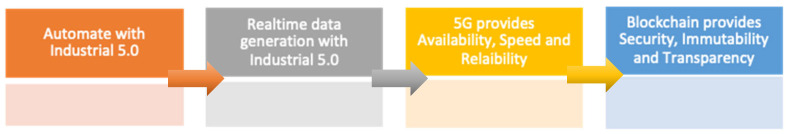
Blockchain implementation roadmap.

**Figure 4 sensors-23-09174-f004:**
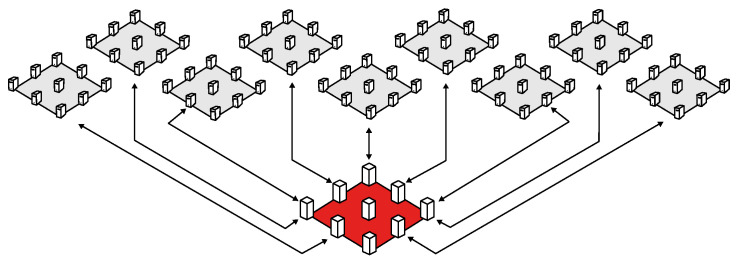
dPoSec blockchain networking.

**Figure 5 sensors-23-09174-f005:**
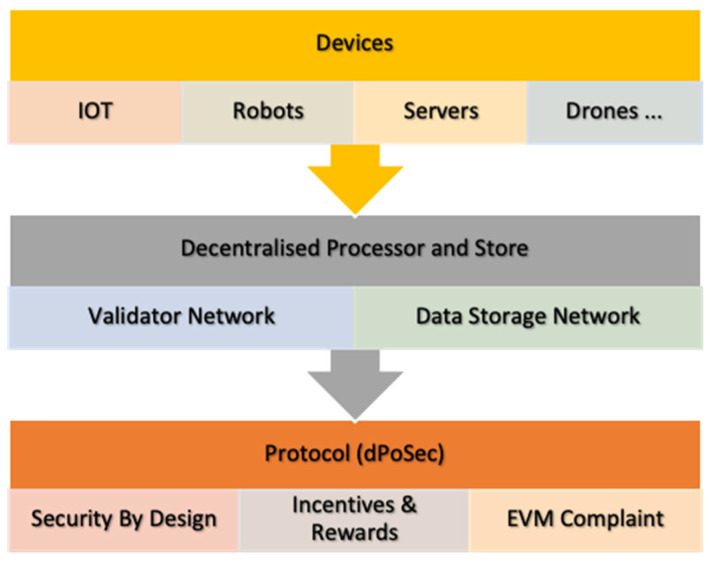
dPoSec blockchain upload process.

**Table 1 sensors-23-09174-t001:** Comparison of consensus mechanisms.

	Proof of Work (PoW)	Proof of Stake (PoS)	BFT Family (BFT)	Distributed Proof of Security (dPoSec)	Proof of Elapsed Time (PoET)
Public	Yes	Yes	Yes	Yes	Yes
Transaction Finality	Eventually	Immediate	Immediate	Immediate	Immediate
Permissionless	Yes	No	No	No	Yes
Token	Generally used to reward participating nodes	Used to elect delegates	Not typically used	Generally used to reward participating nodes	Not typically used
Scalability	Low	High	Low	High	High
Security	High	High	High	High	High
Speed	Slow	Fast	Fast	Fast	Fast
Costs of Participation	High (electricity)	Low	Medium	Medium	Low
Energy Efficiency	Low	High	High	Medium	High
Trustiness	High	Medium	High	High	High
Resilience	High	High	High	High	High
Decentralization	High	Low	Low	Medium	High
Tolerance to Validation	Medium	High	High	Medium	Medium
Blockchain Trilemma	Weak in terms of scalability and energy efficiency	Stronger in terms of scalability and speed than PoW and PoS but weaker in terms of	Stronger in terms of security and speed than PoW and PoS but weaker in terms of scalability and decentralization	Varies based on implementation	Stronger in terms of energy efficiency and decentralization than PoW and PoS but weaker in terms of scalability
Layer 1/Layer 2	Layer 1	Layer 1	Layer 1	Layer 2	Layer 2

## Data Availability

Data are contained within the links: https://naorisprotocol.com/, https://medium.com/@NaorisProtocol, https://www.linkedin.com/company/naorisprotocol, accessed on 14 August 2023.

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
