# Peer review of "Blockchain Protocols and Edge Computing Targeting Industry 5.0 Needs"

_sensors, 2023, doi:10.3390/s23229174_

Round 1

Reviewer 1 Report (Previous Reviewer 2)

Comments and Suggestions for Authors

Good paper.

Author Response

Dear Sir,

Thank you for your review.

Sincerely,

Reviewer 2 Report (New Reviewer)

Comments and Suggestions for Authors

The work entitled “Blockchain Protocols & Edge Computing Targeting Industry 5.0 Needs” was revised. This deals with the adoption of blockchain technology at Industry 5.0. The work presents an interesting landscape analysis with relevant comments and a characteristics overview. However, it fails to introduce the reader to the main topic and presents a clear goal. It is missing a direction. Also, some interesting works deal with Blockchain failures. These should be included.

Part of the text could also be improved. Here are some issues with the current revision

1.       Please revise the entire text formatting.

2.       Please revise: “The vast majority of the work that humans were once responsible for in each, and every industrial physical process can now be done by sensors, which are both more accurate and less expensive (Figure 1).”;

3.       The following sentence, presented on page 3 is repeated

a.       “In the previous chart is presented data from 2019 because it is the most accurate stats that are available since the pandemic crisis”.

4.       Please revise “Companies and governmental entities will spend $10.5 trillion yearly by 2025, up from $3 trillion USD in 2015 [9], where the global spending on cyber- security products and services will be $1.75 trillion cumulatively for the five-year period from 2021 to 2025 [10].”.

5.       PoW should be presented when first appears in the text.

a.       PoW is a consensus algorithm used to secure blockchain networks and validate transactions. Proof of Work (PoW)…

6.       There is a repeated sentence:

a.       The average amount of work needed grows by a factor of the number of zero bits needed, and this can be checked by running a single hash. The average amount of work needed grows by a factor of the number of zero bits needed, and this can be checked by running a single hash.

7.       Pag 8, Istanbul BFT, Delegated Byzantine Fault Tolerance or dBFT, etc. Please include references providing further details for each.

8. On pag 11, where is reference 55? “There are currently more than 70+ different consensus algorithms that support either public or private blockchain networks, or both, and the list is constantly growing [55].”

9.       Chapter 5 should be merged with 6

10.   Please revise “How does the utilization of dPoSec in Industry 5.0 enhance the security posture of the blockchain network, allowing IoT devices to operate with greater trust and efficiency?”

11.   Please revise the paragraphs in section 6

12.   The text after Figure 3 has lengthy phrases, please revise them.

Comments on the Quality of English Language

English language requiring some revision

Author Response

Dear Reviewer,

Thank you for your concerns and care. We have brushed in yellow all changes.

Responding to your aims:

  1.       Please revise the entire text formatting.
    done
    2.       Please revise: “The vast majority of the work that humans were once responsible for in each, and every industrial physical process can now be done by sensors, which are both more accurate and less expensive (Figure 1).”;

Done

3.       The following sentence, presented on page 3 is repeated

Deleted

a.    “In the previous chart is presented data from 2019 because it is the most accurate stats that are available since the pandemic crisis”.
b.    
4.       Please revise “Companies and governmental entities will spend $10.5 trillion yearly by 2025, up from $3 trillion USD in 2015 [9], where the global spending on cyber- security products and services will be $1.75 trillion cumulatively for the five-year period from 2021 to 2025 [10].”.

Revised

5.       PoW should be presented when first appears in the text.

a.    PoW is a consensus algorithm used to secure blockchain networks and validate transactions. Proof of Work (PoW)…

revised

6.       There is a repeated sentence:
a.       The average amount of work needed grows by a factor of the number of zero bits needed, and this can be checked by running a single hash. The average amount of work needed grows by a factor of the number of zero bits needed, and this can be checked by running a single hash.

deleted

7.       Pag 8, Istanbul BFT, Delegated Byzantine Fault Tolerance or dBFT, etc. Please include references providing further details for each.

References were added

8. On pag 11, where is reference 55? “There are currently more than 70+ different consensus algorithms that support either public or private blockchain networks, or both, and the list is constantly growing [55].”

Corrected reference

9.       Chapter 5 should be merged with 6

Indeed it should be, although prior revisor asked to split it. But I couldn’t agree more with you.

10.   Please revise “How does the utilization of dPoSec in Industry 5.0 enhance the security posture of the blockchain network, allowing IoT devices to operate with greater trust and efficiency?”

11.   Please revise the paragraphs in section 6

12.   The text after Figure 3 has lengthy phrases, please revise them.

Revised. It was added and explanatory setence, regarding previous reviewer demands.

Reviewer 3 Report (New Reviewer)

Comments and Suggestions for Authors

The paper offers some interesting insights into the potential of blockchain adoption in smart city and industry 5.0 contexts.

The paper offers some very revealing and novel prespectives at times, but it is this reviewer opinion that it is somehow uneven and that a review would improve the readibility and the homogeneity of the work, thus bringing more value to the important contributions by the authors and also adding more value to the readership  of the journal and boht the industry and the scientific community.   

Also, there are a few instances where the writing style uses and overstatement that blur the powerful insights that could be extracted with the a more sober and objective use of terms and concepts.

Some minor suggestions include: 

Abstract Clarity: The abstract is somewhat lengthy and could benefit from concise summarization. A clearer and more concise abstract would make it easier for readers to grasp the main points of the paper quickly. Also, some points can move to introduction.

 Organization and Flow: The introduction covers a broad range of topics, from 5G networks to sensors and data integrity, without a clear sub-section structure. This may make it challenging for readers to follow the logical flow of ideas. Structuring the content more clearly would enhance readability.

Formatting and Consistency: The section would benefit from improved formatting and consistency in terms of subheadings and numbering. It could be structured more uniformly for easier readability. Some sub-sections are unevenly long as compared to others that are much shorter. It would perhaps be advisable to consider a division of longer sub-sections.

3.4. Byzantine Fault Tolerance Family (BFT) generally known as PoA. The section contains multiple instances of repetition, where similar information is presented, multiple times using different wording. For example, it repeatedly discusses the concepts of safety, liveness, and fault tolerance without adding significant new information. The section is an example of a lengthy text detailing various BFT algorithms. The authors may consider a more concise and focused explanation, highlighting key differentiators and practical implications. Some parts of the section are vague and could offer deeper insights. For instance, when discussing security issues, it mentions that there are "problems with security, integrity, and privacy" related to DiemBFT. Can this be more elaborate or concrete examples be provided. This would perhaps help clarify the reader, on the one hand, and allow for a shorter explanation of the concepts, on the other.

3.5 Distributed Proof of Security (dPoSec): It would perhaps be sound to provide definitions and a solid explanation of the scientific basis of the concept and some of the affirmations/opinions presented. At times, the writing style can be equated with too casual or opinative , e.g.:

  1. "a significant step forward in the field of blockchain consensus algorithms."
  2. "a well-rounded and versatile platform."
  3. "an ideal solution for any situation."
  4. "extremely scalable."
  5. "sophisticated blockchain primitives."
  6. "full control over their data and giving them the ability to make informed decisions."
  7. "one-of-a-kind validator selection process."
  8. "a more secure and reliable consensus mechanism."
  9. "utilization of cryptographic signatures helps to ensure that the network's integrity is maintained even in the event that one or more of its nodes are breached."
  10. "extremely adaptable."
  11. "can contribute significantly to Industry 5.0."
  12. "a reliable and effective consensus mechanism."

At other points, the ideas conveyed in the text could be more detailed for the readershop that might not be familiar with the concepts presented. For example:

  1. It incorporates advanced security measures that are based on trust establishment among nodes to ensure a fast, scalable, and secure network operation.” What are these advanced security measures?
    “represents a significant step forward in the evolution of consensus algorithms.” Why is this a step forward? In relation to what?
  2. “dPoSec is a well-rounded and versatile platform”. Is it a consensus algorithm or a "versatile platform"?  
  3. The algorithm was developed to be extremely scalable, and its architecture makes use of sophisticated blockchain primitives to improve both its performance and its efficiency.” It would be interesting for the readership to learn more about this is done.
  4. dPoSec's blockchain primitives are designed to be more efficient than those of other blockchains, which results in a number of benefits, including a reduction in the amount of time needed to process transactions and an increase in scalability. Because dPoSec is an on-demand privacy platform, users are able to protect the confidentiality of their data and take advantage of the privacy benefits that come along with using the service. This means that users can choose which information they want to share and with whom, giving them full control over their data and giving them the ability to make informed decisions. The peer-to-peer discovery feature of the platform makes it easy for nodes to find each other and establish connections, thereby lowering the network's latency and increasing its overall efficiency.” E.g. what is the estimated amount of time reduction? Why and how is this achieved? This are the type of questions that would interest the reader.
  5. How to measure: “their reputation, and their performance.”
  6. “because it uses advanced security measures to increase trust in the running nodes” What are these advanced security measures? Advanced compared to what?
  7. “dPoSec was developed with a particular emphasis on security, efficiency, and scalability in order to cater to the requirements of a wide range of sectors and assist those sectors in maximizing the potential of blockchain technology. The algorithm is extremely adaptable and can be tailored to the particular requirements of each sector, which enables it to function as a solution that is both versatile and scalable. For instance, it can be used in the financial sector to ensure the safety and efficacy of payment transactions, and it can also be used in the healthcare sector to secure and manage electronic medical records. Both of these applications can be found in the financial sector. It is designed to meet the needs of a variety of industries and to assist those industries in making use of the benefits offered by blockchain technology. It accomplishes this by combining advanced security measures with a focus on scalability [21].” Is this a repetition of previous text?

 Section 3.5 seems to repeats certain points and phrases, such as describing dPoSec as "a significant step forward in the field of blockchain consensus algorithms" multiple times. This type of repetitions decrease the impact of the text on the reader due to lack of conciseness and adds little new value or substantial content. This same section mentions that dPoSec combines aspects of Proof of Stake (PoS) and Byzantine Fault Tolerance (BFT) protocols, but it does not provide specific details about how this combination works or what distinguishes dPoSec from other consensus mechanisms. It lacks a clear explanation of the technical aspects. It somehow generalizes the applicability of dPoSec without providing tailored insights for different use cases. It would be interesting for the reader to have more details or references to complementary reading,

 3.6: “PoET is designed to address the problem of energy waste in Proof of Work (PoW) and Proof of Stake (PoS) systems.” and in 3.2 “PoS is better for the environment and uses less energy than PoW because it doesn't require miners to solve hard math problems to verify transactions.” This two different sentences seem conflicting, as “ Proof of Stake (PoS) is a consensus mechanism that requires participants to put cryptocurrency as collateral for the opportunity to successfully approve transactions1It is considered more energy-efficient than Proof of Work (PoW), which is used by Bitcoin and consumes over 99% more energy than PoS networks like Tezos, Polkadot, or Solana2Ethereum, for instance, has transitioned from PoW to PoS, making it a green blockchain3The energy consumption of Ethereum’s PoS mechanism is approximately ~0.0026 TWh/yr across the entire global network3To put this into perspective, the annualized energy consumption of PoS Ethereum is significantly lower than that of various industries such as global data centers, gold mining, Bitcoin, and even Netflix3.”

 Table 1 : The term "Low", "High", "Slow", "Fast", and so on at table is not comparable. A further detailed of the meaning of these terms or their quantification would improve the value of the paper. E.g, Scalability: Measured in transactions per second (TPS) or operations per second (OPS). Transaction Finality: Measured in seconds (s) for the time it takes for a transaction to be considered final. Speed: Measured in seconds or milliseconds (ms) for transaction confirmation time or block creation time. Energy Efficiency: Measured in units of energy, such as kilowatt-hours (kWh) for the energy consumption required by the consensus mechanism. also add: Cross-Chain Compatibility, Smart Contract Support , Governance Mechanisms , Privacy and Anonymity , Data Throughput (Data Volume), Sybil Attack Resistance,  Governance Token Metrics , Transaction Privacy , Legal and Regulatory Compliance , Stability and Consistency ,

Discussion: There is some repetition of ideas, particularly when emphasizing the benefits of dPoSec. Consider consolidating similar points to make the text more concise.

In conclusion, the paper is very interesting but could work better if a certain excess of repetition of ideas and concepts from earlier sections were avoided. While the conclusions mention the potential applications of the discussed technologies, such as smart factories and self-driving cars, further insights into concrete real-world examples or case studies could make the benefits more tangible and relatable. The document discusses the disruptive potential of the integrated technologies, it could benefit from a forward-looking perspective, highlighting potential future developments and trends in Industry 5.0 and technology integration.

Also, a full revision to harmonize the style would be advised. The text up to 3.4 language is more concise and  scientific. There seems to be an abrupt change after 3.5, with a less sober and objective style and a more commercial tone.

Some sentences are quite lengthy. Simplifying sentence structure and using more concise language could improve readability.

 Except for figure 1, other figures and tables are not co-ref in the text before its appearance. These cases need to be revised. Figures must always be properly mention in advance. There is a figure without caption. Also, figures provide visual information, but the captions and descriptions could at cases be more informative.

Comments on the Quality of English Language

See comments above.

Author Response

Dear Reviewer,

Thank you for your care and concerns. We have marked all your converns in yellow.

Response:

1) Abstract Clarity: The abstract is somewhat lengthy and could benefit from concise summarization. A clearer and more concise abstract would make it easier for readers to grasp the main points of the paper quickly. Also, some points can move to introduction.

The abstract was smaller, allthough prior revisor asked for more detailed abstract.

2) Organization and Flow: The introduction covers a broad range of topics, from 5G networks to sensors and data integrity, without a clear sub-section structure. This may make it challenging for readers to follow the logical flow of ideas. Structuring the content more clearly would enhance readability.

3) Formatting and Consistency: The section would benefit from improved formatting and consistency in terms of subheadings and numbering. It could be structured more uniformly for easier readability. Some sub-sections are unevenly long as compared to others that are much shorter. It would perhaps be advisable to consider a division of longer sub-sections.

It was made subsections in the past but other reviewers asked to avoid framing the document more than waht was given to the template

4) 3.4. Byzantine Fault Tolerance Family (BFT) generally known as PoA. The section contains multiple instances of repetition, where similar information is presented, multiple times using different wording. For example, it repeatedly discusses the concepts of safety, liveness, and fault tolerance without adding significant new information. The section is an example of a lengthy text detailing various BFT algorithms. The authors may consider a more concise and focused explanation, highlighting key differentiators and practical implications. Some parts of the section are vague and could offer deeper insights. For instance, when discussing security issues, it mentions that there are "problems with security, integrity, and privacy" related to DiemBFT. Can this be more elaborate or concrete examples be provided. This would perhaps help clarify the reader, on the one hand, and allow for a shorter explanation of the concepts, on the other.

At first, one standard exemple of BFT and one dBFT was given. Reviewers asked to detail and reference all the BFT’s

5) 3.5 Distributed Proof of Security (dPoSec): It would perhaps be sound to provide definitions and a solid explanation of the scientific basis of the concept and some of the affirmations/opinions presented. At times, the writing style can be equated with too casual or opinative , e.g.:

All setences were reviewed

6) At other points, the ideas conveyed in the text could be more detailed for the readershop that might not be familiar with the concepts presented. For example:

“It incorporates advanced security measures that are based on trust establishment among nodes to ensure a fast, scalable, and secure network operation.” What are these advanced security measures?

Reviewed

“dPoSec is a well-rounded and versatile platform”. Is it a consensus algorithm or a "versatile platform"?  

Reviewed

“The algorithm was developed to be extremely scalable, and its architecture makes use of sophisticated blockchain primitives to improve both its performance and its efficiency.” It would be interesting for the readership to learn more about this is done.”

We could have done more description about the primitives although prior reviewers said tha the document was focusing too much on blockchain architecture instead of the aim of abstract.

“dPoSec's blockchain primitives are designed to be more efficient than those of other blockchains, which results in a number of benefits, including a reduction in the amount of time needed to process transactions and an increase in scalability. Because dPoSec is an on-demand privacy platform, users are able to protect the confidentiality of their data and take advantage of the privacy benefits that come along with using the service. This means that users can choose which information they want to share and with whom, giving them full control over their data and giving them the ability to make informed decisions. The peer-to-peer discovery feature of the platform makes it easy for nodes to find each other and establish connections, thereby lowering the network's latency and increasing its overall efficiency.” E.g. what is the estimated amount of time reduction? Why and how is this achieved? This are the type of questions that would interest the reader.

Reviewed

“because it uses advanced security measures to increase trust in the running nodes” What are these advanced security measures? Advanced compared to what?

They are described in the same paragraph and also detailed in your previews requests

“dPoSec was developed with a particular emphasis on security, efficiency, and scalability in order to cater to the requirements of a wide range of sectors and assist those sectors in maximizing the potential of blockchain technology. The algorithm is extremely adaptable and can be tailored to the particular requirements of each sector, which enables it to function as a solution that is both versatile and scalable. For instance, it can be used in the financial sector to ensure the safety and efficacy of payment transactions, and it can also be used in the healthcare sector to secure and manage electronic medical records. Both of these applications can be found in the financial sector. It is designed to meet the needs of a variety of industries and to assist those industries in making use of the benefits offered by blockchain technology. It accomplishes this by combining advanced security measures with a focus on scalability [21].” Is this a repetition of previous text?

Prior reviewers asked for exemples.

 Section 3.5 seems to repeats certain points and phrases, such as describing dPoSec as "a significant step forward in the field of blockchain consensus algorithms" multiple times. This type of repetitions decrease the impact of the text on the reader due to lack of conciseness and adds little new value or substantial content. This same section mentions that dPoSec combines aspects of Proof of Stake (PoS) and Byzantine Fault Tolerance (BFT) protocols, but it does not provide specific details about how this combination works or what distinguishes dPoSec from other consensus mechanisms. It lacks a clear explanation of the technical aspects. It somehow generalizes the applicability of dPoSec without providing tailored insights for different use cases. It would be interesting for the reader to have more details or references to complementary reading,

It inherits aspects from PoS and BFT in a new consensos

“ 3.6: “PoET is designed to address the problem of energy waste in Proof of Work (PoW) and Proof of Stake (PoS) systems.” and in 3.2 “PoS is better for the environment and uses less energy than PoW because it doesn't require miners to solve hard math problems to verify transactions.” 
This two different sentences seem conflicting, as “ Proof of Stake (PoS) is a consensus mechanism that requires participants to put cryptocurrency as collateral for the opportunity to successfully approve transactions1. It is considered more energy-efficient than Proof of Work (PoW), which is used by Bitcoin and consumes over 99% more energy than PoS networks like Tezos, Polkadot, or Solana2. 
Ethereum, for instance, has transitioned from PoW to PoS, making it a green blockchain3. The energy consumption of Ethereum’s PoS mechanism is approximately ~0.0026 TWh/yr across the entire global network3. To put this into perspective, the annualized energy consumption of PoS Ethereum is significantly lower than that of various industries such as global data centers, gold mining, Bitcoin, and even Netflix3.””

PoW (low energy efficiency) PoS (high energy efficiency)
PoW consumes more energy. 

Eventhough, PoET consumes even less energy.

“ Except for figure 1, other figures and tables are not co-ref in the text before its appearance. These cases need to be revised. Figures must always be properly mention in advance. There is a figure without caption. Also, figures provide visual information, but the captions and descriptions could at cases be more informative.”

Reviewed.

Round 2

Reviewer 2 Report (New Reviewer)

Comments and Suggestions for Authors

 No comments

This manuscript is a resubmission of an earlier submission. The following is a list of the peer review reports and author responses from that submission.

Round 1

Reviewer 1 Report

Comments and Suggestions for Authors

Dear Authors, 

Thank you for your effort to address previous reviewers' concerns. In this revision you have tried to explain what your objective is, but I would suggest that you dial down the subjectivity of the quality of the paper and focus on the problems you are trying to address. 

As one of the previous reviewers suggested, and your added objectives in the introduction indicate, this manuscript seems to be a systematic review of blockchain protocols and possibilities to adopt blockchain for developing solution for Smart City, Smart Home, Healthcare, Smart Agriculture, Autonomous Vehicles, and Supply Chain Management within Industry 5.0. However, in order to make this a review, please, follow the guidelines e.g. from: http://prisma-statement.org/prismastatement/checklist.aspx

In the added material you mention DPOS as a suggested solution for security issues in Industry 5.0. However, the reference that you provided indicates that one of the biggest flaws of DPOS is security.

The last paragraph ends abruptly and conclusion section looks like discussion that you wanted to provide before concluding your manuscript.

Comments on the Quality of English Language

I didn't find problems with English language. 

Author Response

Dear Reviewer,

We will responde below and upload the latest version in this submission. Thank you for your valuable comments.

Thank you for your effort to address previous reviewers' concerns. In this revision you have tried to explain what your objective is, but I would suggest that you dial down the subjectivity of the quality of the paper and focus on the problems you are trying to address. 

The authors thank the reviewer’s comment that will allow a better explanation of the main added value of this paper.

Along the document, we had the effort to present the technology “assets” (like blockchain, edge computing, sensors, transfer data comms) needed to power industry challenges, regarding rapidness, security, data integrity and cohesion. In the least sections we present an architecture that gather those technologies that should been implemented on the shop floor.

The performed changes are indicated in yellow colour in the revised version of the paper.

As one of the previous reviewers suggested, and your added objectives in the introduction indicate, this manuscript seems to be a systematic review of blockchain protocols and possibilities to adopt blockchain for developing solution for Smart City, Smart Home, Healthcare, Smart Agriculture, Autonomous Vehicles, and Supply Chain Management within Industry 5.0. However, in order to make this a review, please, follow the guidelines e.g. from: http://prisma-statement.org/prismastatement/checklist.aspx

 As explained in the abstract, suggested by previous reviewers, it was not our intention  to make a review about blockchain protocols, but, in order to address several industry issues, properties, features and characteristics needed to be reviewed.

In the added material you mention DPOS as a suggested solution for security issues in Industry 5.0. However, the reference that you provided indicates that one of the biggest flaws of DPOS is security.

In our research we believe dPoSec is one of the safest blockchain protocol, beyond other advantages, that can be consulted from reference 21 (https://docsend.com/view/piiy2cvzghx262ma)

The authors thank the reviewer’s comment that will allow a better explanation of the main added value of this paper. Effectively, the PoS protocol (3.2) has flaws, being mitigated with DPoS and, particularly, with dPoSec protocol.

The last paragraph ends abruptly and conclusion section looks like discussion that you wanted to provide before concluding your manuscript.

The authors agree with reviewer comment. It is a good observation that allowed significant improvement of the paper. It is an ongoing process, merging blockchain and edge computing, and also a requirement for this from previous reviewers.

The performed changes are indicated in yellow colour in the revised version of the paper, where we emphasise the dPoSec protocol in terms of industry safety for distributed data processing.

Reviewer 2 Report

Comments and Suggestions for Authors

This paper discussed blockchain protocols & edge computing targeting industry 5.0 needs. The topic is timely. This paper is well organized. A minor revision is suggested for this paper.

1.         The abstract section should be simplified.

2.         Different from the technology-driven I4.0, I5.0 is value-oriented. The authors are suggested to discuss how to re-organize the technologies for adding value.

3.         Sustainability considers three pillars, namely, economy, environment, and society. I5.0 emphasizes more on the environment and society. The authors are suggested to enhance the discussions on the society and environment aspects of blockchain protocols & edge computing technology.

4.         Resilience is one of the three propositions of I5.0 blueprint. The authors are suggested to enhance the discussions on resilience. Resilience refers to the ability of a system to keep or recover quickly to a stable state during and after a major mishap or in the presence of continuous significant stresses. It could be enhanced by modularized/open architecture design, robust control, and predictive decision techniques. For instance, secure blockchain middleware for decentralized iiot; industry 5.0 prospect and retrospect; blockchained smart contract pyramid-driven multi-agent autonomous process control for resilient individualised manufacturing; blockchained smart contract system as the digital twin of decentralized autonomous manufacturing towards resilience. You may present more discussions on this organizational resilience aspect of computing technology.

Author Response

Dear Reviewer,

We will responde below and upload the latest version in this submission. Thank you for your valuable comments.

The abstract section should be simplified.

The authors agree with reviewer comment. The abstract section was reviewed in order accomplished previous reviewer comments, like providing the arguments more clearly.

Different from the technology-driven I4.0, I5.0 is value-oriented. The authors are suggested to discuss how to re-organize the technologies for adding value.

Thank you for this suggestion. It is an ongoing issue that should be addressed in future, when architecture and distributed system is consolidated.

Sustainability considers three pillars, namely, economy, environment, and society. I5.0 emphasizes more on the environment and society. The authors are suggested to enhance the discussions on the society and environment aspects of blockchain protocols & edge computing technology.

Thank you for this suggestion. Somehow, blockchain systems consume large amounts of energy and that is one of the concerns in our assessment of blockchain protocols. Currently we have no more data about the gather of blockchain and edge computing. But it is an issue to have in account for further research.

Resilience is one of the three propositions of I5.0 blueprint. The authors are suggested to enhance the discussions on resilience. Resilience refers to the ability of a system to keep or recover quickly to a stable state during and after a major mishap or in the presence of continuous significant stresses. It could be enhanced by modularized/open architecture design, robust control, and predictive decision techniques. For instance, secure blockchain middleware for decentralized iiot; industry 5.0 prospect and retrospect; blockchained smart contract pyramid-driven multi-agent autonomous process control for resilient individualised manufacturing; blockchained smart contract system as the digital twin of decentralized autonomous manufacturing towards resilience. You may present more discussions on this organizational resilience aspect of computing technology.

It is a great suggestion indeed. In fact resilience is one of our concerns, specially when we are doing tests and experiments in sensors, injecting blockchain firmware and at the same time edge-processing data.

The performed changes are indicated in yellow colour in the revised version of the paper, where we emphasise the dPoSec protocol in terms of industry safety for distributed data processing.

Round 2

Reviewer 1 Report

Comments and Suggestions for Authors

Dear Authors,

Thank you for your explanations that followed the changes in the manuscript. I think that you are on the right track, I now see the great potential and how much effort you have invested in your work but I think that you still didn't understand the comment to look at systematic literature review guidelines.  Your claim "we had the effort to present the technology “assets” (like blockchain, edge computing, sensors, transfer data comms) needed to power industry challenges, regarding rapidness, security, data integrity and cohesion." further exacerbates your case and makes the need for systematic literature review even stronger. I strongly believe that if you follow the guidelines of he systematic literature review mapping (one example is http://www.prisma-statement.org/) and frame your manuscript that way (having clear hypotheses, well defined search for the technology assets as you call them, well defined questions that you want to answer) you will significantly improve the manuscript. 

Comments on the Quality of English Language

English needs improvement, even though I understand clearly what authors write about. 

Author Response

Dear Sir,

We have edited and corrected your considerations and notes and included more results regarding your reviews.

In first hand, thanks for your comments in our document. We’ve taken carefully your comments in order to improve our research and work.

We’ve have improved introduction regarding the relevance of 5g, iiot and sensors for blockchain and edge computing research.

We also have addressed the issues regarding clear hypothesis, well defined questions and defined search for the technology, having has guideline the prisma statement.

Thank you very much for your care about this ongoing research involving one major blockchain player and device manufacturers.

Sincerely,

August 2nd 2023

Miguel Oliveira
Sumit Chauchan
Filipe Pereira
Manuel Felgueiras
David Carvalho
